

# Resistance of mature and elderly bodybuilders to anaerobic energy supply load

Andrii Chernozub[1,2], Alla Aloshyna[1], Georgiy Korobeynikov[3,4,5], Vadym Koval[6], Yurii Havrylov[1], Liudmyla Sherstiuk[7], Vladimir Potop[8,9], Consuela Andreea Timnea-Florescu[10] and Olivia Carmen Timnea[11]

[1] Lesya Ukrainka Volyn National University, Lutsk, Ukraine
[2] The Scientific Research Center of Modern Kinesiology, Ujhorod, Ukraine
[3] National University of Ukraine on Physical Education and Sport, Kyiv, Ukraine
[4] Institute of Psychology, German Sport University Cologne, Cologne, Germany
[5] Uzbek State University of Physical Education and Sports, Chirchiq, Tashkent, Uzbekistan
[6] Academician Stepan Demianchuk International University of Economics and Humanities, Rivne, Ukraine
[7] Petro Mohyla Black Sea National University, Mykolaiv, Ukraine
[8] Institute of Phisical Education and Sport, Moldova State University, Chisinau, Moldova
[9] Department of Physical Education and Sports, National University of Science and Technology Politehnica Bucharest, University Center Pitesti, Pitesti, Romania
[10] Chiajna Medical Center, Chiajna, Romania
[11] Romanian-American University, Bucharest, Romania

Corresponding author
Vladimir Potop,
vladimir_potop@yahoo.com

## ABSTRACT

**Background**. This study aims to assess the resistance level of beginner bodybuilders of mature and elderly age to strength training performed under anaerobic energy supply modes (creatine phosphokinase and glycolysis).

**Methods**. The study involved 60 men who had been engaged in bodybuilding for only 12 months. Group A included men aged $24 \pm 1.5$ years (early mature age), group B consisted of men aged $40 \pm 2.1$ years (late mature age), and group C comprised men aged $58 \pm 1.6$ years (elderly). An integral method of quantitative assessment of power loads was used to determine 1 repetitions maximum (RM) and load parameters for each anaerobic energy supply mode. Resistance levels were evaluated based on adaptive-compensatory responses to training stimuli, using heart rate variability (HRV) and biochemical blood markers (creatine phosphokinase (CPK), lactate dehydrogenase (LDH), testosterone, cortisol).

**Results**. Pre-exercise results showed that participants in groups A and C exhibited more tense heart rhythm regulation, with autonomic balance shifted toward sympathetic dominance. Group B showed creatine phosphokinase activity in the blood that was twice as high as in the other groups. Cortisol levels in group C were on average 17.6% higher than in groups A and B. Post-exercise results under the creatine phosphokinase energy supply mode revealed increased autonomic regulation and decreased sympathetic tone in groups B and C. In contrast, Group A showed heightened heart rate regulation tension (standard deviation of RR intervals (SDNN) $-38.5\%$) and increased central sinus rhythm regulation (very low frequency (VLF) $+34.3\%$). In groups B and C, creatine phosphate was the primary energy source, while in group A, due to insufficient creatine phosphate reserves, muscle glycogen was additionally utilized, leading to increased blood LDH ($+94.7\%$) and cortisol ($+133.2\%$). Under

the anaerobic glycolysis mode, groups A and B showed increased vagal influence on the sinus node, while group C exhibited increased sympathetic tone (low frequency (LF) +11.4%) and reduced autonomic regulation (high frequency (HF) −5.9%). In group A, only LDH activity increased (+90.2%) compared to rest. Cortisol levels decreased significantly in groups A (−21.0%) and C (−12.4%), indicating activation of compensatory mechanisms.

**Conclusions**. The findings underscore the need to develop tailored load adjustment strategies for mature and elderly individuals beginning bodybuilding. Load regimes should consider individual strength capacities and adaptive reserves. The use of heart rate variability (HRV) and blood biochemical markers is validated as an informative approach for assessing resistance to training-induced stress.

## INTRODUCTION

With the increasing popularity of various bodybuilding class models among different age groups, the challenge arises of identifying effective methods to ensure that the intensity of workouts aligns with individuals' functional capacities. Assessing the resistance level to such stressful stimuli is important for young, mature, and elderly people (*Lin et al., 2022*; *Baxter et al., 2024*; *Gambassi et al., 2024*). The study of adaptation in bodybuilding has been of interest to scientists in various fields for many years. However, many works are devoted to studying the adaptation and compensatory reactions of young beginners or elite athletes (*Chernozub et al., 2023*; *Mantri et al., 2023*). Adaptation-compensatory reactions reflect the body's ability to counteract acute stress stimuli by activating short-term adaptation processes or compensatory responses. Relevant studies in most cases are aimed at optimizing the training, developing innovative programs, and studying the effectiveness of using ergogenic means (*Schoenfeld et al., 2023*). Much attention is paid to the study of issues related to the restoration of adaptive reserves of people after the end of their sports career using a wide range of methods and principles of bodybuilding (*Rukstela et al., 2023*). At the same time, the study of adaptative and compensatory reactions of the people, who began training in bodybuilding at a mature or elderly age, has not been paid attention to by scientists. The complexity of this problem lies in the absence of a system for assessing the body resistance of these age categories to bodybuilding loads. There is a lack of clear patterns of change in the relevant markers for assessing adaptive reserves in these categories, depending on the load modes and the energy supply mechanisms involved. Moreover, there is no scientific substantiation of choosing an effective set of physiological and biochemical methods for controlling load regimes for people with different levels of resistance to stressful stimuli (*Grässler et al., 2021*; *Morcillo-Losa et al., 2024*).

In modern bodybuilding, using various training models in combination with different load modes is considered effective in achieving success (*Lopez et al., 2021*; *He et al., 2024*). However, only a few researchers (*Chernozub et al., 2020*; *Barbosa et al., 2024*) highlight the

issue of the absence of an efficient system for evaluating adaptive reserve levels. The lack of an integrated approach to controlling compensatory reactions is due to inadequate loads on the body's functional reserves. Implementing these issues will allow timely detection of adaptation failure manifestations and influence the correction of exercise regimes and training models. This problem is especially acute for people who take up bodybuilding at a mature and elderly age.

**Purpose of the study**. To define the resistance level of beginner bodybuilders in mature and elderly age to strength training in anaerobic energy supply modes (creatine phosphokinase and glycolysis).

## MATERIALS & METHODS

### Participants

The study involved 60 men of different age groups who had started bodybuilding training only 12 months prior. Three experimental groups (A, B, C) were formed. The study participants had no contraindications to anaerobic exercise based on the results of a comprehensive medical and biological evaluation. Group A involved 20 people aged $24 \pm 1.5$ years (early mature age). Group B included 20 men of late mature age ($40 \pm 2.1$ years). Group C had 20 elderly men aged $58 \pm 1.6$ years. The study was conducted in 2024 at the Research Centre for Modern Kinesiology "KINEZUS" and its branches (Mykolaiv, Odesa, Chernivtsi, Rivne, Ukraine). The algorithm, structure, and methods of the study were approved by the ethical committee for biomedical research of Lesya Ukrainka Volyn National University (Minutes No. 023384 dated September 19, 2024). The study participants were explained the risks and benefits and signed an informed consent form, which was prepared following the ethical standards of the Declaration of Helsinki.

### Measurements

#### Maximum muscle strength

The assessment of the maximum strength development of the muscle groups (1RM) among the participants of the examined groups was carried out at the beginning of the study. Taking into account the initial level of training of participants and their age and physiological characteristics of the organism simulator exercises were used to monitor the 1RM indicator. Because of the age of the study participants, a safe exercise technique was used. The exercise 'hammer chest press' was used to determine the maximum strength of the pectoral muscles. The control exercise 'hammer lat pulldown' was used to control the maximum strength of the back muscles. The development of the maximum strength of the triceps brachii muscle was assessed using the exercise 'hammer triceps extension'. 1RM parameters determination in each control exercise was carried out according to the generally accepted bodybuilding method.

#### Parameters of power loads

Using the integral method of assessing power loads (*Chernozub et al., 2018*), the load factor (Ra) was determined for each anaerobic mode of energy supply (creatine phosphokinase

and glycolytic). Using 1 RM and Ra results, the projectile working mass (m) and the load volume (Wn) in a set were calculated. The load duration in a set in the creatine phosphokinase energy supply mode, until the complete depletion of creatine phosphate reserves, is no more than 25 s. The exercise duration in a set in the anaerobic glycolysis, until the muscle glycogen reserves are completely depleted, is 60–120 s.

### Heart rate variability

To determine heart rate variability (HRV) indicators (in the time and frequency domains), the Polar V800 heart rate monitor was used. Intervals between heartbeats (RR intervals) were recorded using the H10 sensor. Data processing was carried out using Polar Flow and Kubios HRV Standard 3.5.0. The baseline level of the standard deviation of RR intervals (SDNN, ms) and its changes in response to a stimulus were assessed. Particular attention was given to evaluating the spectral analysis parameters of heart rate. Primarily, the low-frequency (LF, %) and high-frequency (HF, %) indicators were compared to determine autonomic balance parameters (LF/HF). The obtained data allow assessing the level of regulatory stress in heart rhythm among the subjects. Monitoring of the very-low-frequency (VLF, %) component enabled the evaluation of changes in the central regulation of sinus rhythm, indicating either its strengthening or weakening. Before the start of the study, the subjects lay quietly with their eyes closed in a calm environment at a temperature of about 22 °C for 40 min. RR interval signal recordings for each subject were conducted three times at rest while sitting on a bench: first, before physical load; second, after exertion under the creatine-phosphate energy supply mode; and third, after exertion under the anaerobic glycolysis energy supply mode. The optimal duration for RR interval recording, according to HRV research standards (*Jin et al., 2024*), was 5 min.

### Biochemical parameters

Laboratory analysis was conducted to assess the activity of lactate dehydrogenase and creatine phosphokinase enzymes and to determine the testosterone and cortisol concentrations in the subjects' blood. A kinetic method was used to measure the necessary parameters of lactate dehydrogenase (LDH) and creatine phosphokinase (CPK) enzymes. For the practical implementation of this method, laboratory specialists used PRESTIGE 24i LQ LDH reagents in combination with High Technology Inc equipment. The concentration of the glucocorticoid hormone cortisol in the blood of the study participants was determined using the enzyme-linked immunosorbent assay (ELISA). The concentration of this biomarker was measured using the Alkor Bio equipment and Steroid-IFA reagents. A similar laboratory method was applied to determine testosterone concentration parameters in the blood. Special attention was given to the reference values of the monitored biochemical blood indicators, taking into account the age-related characteristics of the subjects. Blood sampling took place in a specially equipped room at the scientific research center of modern kinesiology, "KINEZUS", and was performed by a laboratory medical professional under the supervision of a physician. Internationally accepted standards for conducting such studies were strictly followed. Control was carried out over the baseline levels (before stress stimuli) of the studied biochemical blood indicators and their changes in response to each of the two test tasks (loads). The samples obtained after each blood collection were placed

in special medical freezing containers and delivered to the laboratory. During the study, 180 samples were taken and 720 indicators were obtained for further analysis.

## Experimental design

The study was conducted in several stages throughout 2024.

In the first stage, three resistance exercises using Hammer-strength equipment (bench press, vertical pull, and arm extensions) were selected to achieve the research objectives. Performing these exercises allows for detailed and maximal exhaustion of agonist and synergist muscle groups while minimizing the load on stabilizer muscle groups. This approach reduces the risk of adaptation failure, especially for individuals with low functional fitness levels (beginners). During each control exercise execution, under the specified technique, the maximum strength development (1RM) indicators (repetitions maximum) were determined. At this stage, two control test tasks were developed. The rest between the control tests was 50–60 min to restore energy reserves. The study participants performed a prescribed set of three strength exercises, completing four sets for each exercise in a test. The rest between sets was 60 s, and between exercises, 120 s. In the first control test, the projectile working mass indicators (m) and the load volume in a set (Wn) were calculated for loads (Ra = 0.74) in the creatine phosphokinase energy supply mode. In the second control test, the load indicators (Ra = 0.67) were calculated for the anaerobic glycolysis energy supply mode.

In the second stage, the resistance levels of men (beginner bodybuilders) from different age groups to strength training loads were studied. The changes in HRV parameters and biochemical blood markers were assessed to analyze the adaptive-compensatory responses of their bodies to the loads in the specified control tests. A comparative analysis of these indicators was conducted both at rest and after strength exercises with anaerobic energy supply modes (phosphocreatine and glycolytic). The obtained results were then processed and analyzed.

## Statistical analysis

The IBM SPSS Statistics 27 software package (IBM Corp., Armonk, NY, USA) was used for statistical analysis of the study results. The G-Power 3.1.96 software (Kiel, Germany) allows for determining the smallest sample size for the study (statistical power calculation). With an estimated power of 0.80 and an alpha of 0.05, a total sample of 20 was required to identify a meaningful effect size of 0.25. The median (Me) and interquartile range (IQR) were assessed with non-parametric statistical analysis methods. The Kruskal Wallis H test was used to compare baseline parameters between the three groups of subjects. The Wilcoxon signed-rank test was used to compare two dependent samples with each other (*Nasledov, 2013*). Statistical results are reported according to the APA style (*Yağin et al., 2021*).

## RESULTS

Table 1 presents the results of maximum strength (1 RM) indicators and the main load parameters in the control testing of participants of the examined groups.

**Table 1 Results of the maximum strength (1 RM) indicators and the main load parameters during control tests in the examined groups (median, IQR), $n = 60$.**

| Indicators | Study participants | | |
|---|---|---|---|
| | Group A (early mature age, $24 \pm 1.5$) | Group B (late mature age, $40 \pm 2.1$) | Group C (elderly men, $58 \pm 1.6$) |
| **Maximum strength (1RM, kg)** | | | |
| Hammer chest press | 88.50 (5.42) $H = 43.43; p < 0.001$ | 93.0 (4.50) $H = 43.43; p < 0.001$ | 72.50 (6.75) $H = 43.43; p < 0.001$ |
| Hammer lat pulldown | 68.50 (5.75) $H = 33.53; p < 0.001$ | 72.00 (2.75) $H = 33.53; p < 0.001$ | 65.00 (3.75) $H = 33.53; p < 0.001$ |
| Hammer triceps extension | 42.50 (3.83) $H = 9.88; p = 0.007$ | 43.50 (3.12) $H = 9.88; p = 0.007$ | 41.00 (3.80) $H = 9.88; p = 0.007$ |
| **$m^1$ (working weight of the equipment, kg) in creatine kinase energy supply mode** | | | |
| Hammer chest press | 65.45 (4.22) $H = 43.39; p < 0.001$ | 68.85 (3.53) $H = 43.39; p < 0.001$ | 53.60 (5.10) $H = 43.39; p < 0.001$ |
| Hammer lat pulldown | 50.70 (4.53) $H = 33.78; p < 0.001$ | 53.30 (3.40) $H = 33.78; p < 0.001$ | 48.50 (2.87) $H = 33.78; p < 0.001$ |
| Hammer triceps extension | 31.50 (3.30) $H = 10.83; p = 0.04$ | 32.20 (2.30) $H = 10.83; p = 0.04$ | 30.32 (2.95) $H = 10.83; p = 0.04$ |
| **$Wn^1$ (load volume in a set, kg) in creatine kinase energy supply mode** | | | |
| Chest press hammer | 261.80 (16.20) $H = 43.39; p < 0.001$ | 275.00 (13.46) $H = 43.39; p < 0.001$ | 214.40 (20.00) $H = 43.39; p < 0.001$ |
| Hammer lat pulldown | 202.80 (17.00) $H = 33.78; p < 0.001$ | 213.20 (13.04) $H = 33.78; p < 0.001$ | 194.00 (11.00) $H = 33.78; p < 0.001$ |
| Hammer triceps extension | 126.56 (12.54) $H = 10.83; p = 0.04$ | 128.80 (9.50) $H = 10.83; p = 0.04$ | 123.40 (13.70) $H = 10.83; p = 0.04$ |
| **$m^2$ (working weight of the equipment, kg) in anaerobic glycolysis energy supply mode** | | | |
| Hammer chest press | 59.30 (3.59) $H = 41.35; p < 0.001$ | 62.30 (3.04) $H = 41.35; p < 0.001$ | 48.55 (4.90) $H = 41.35; p < 0.001$ |
| Hammer lat pulldown | 46.00 (3.80) $H = 33.77; p < 0.001$ | 48.20 (2.78) $H = 33.77; p < 0.001$ | 43.85 (2.50) $H = 33.77; p < 0.001$ |
| Hammer triceps extension | 28.50 (2.67) $H = 9.70; p = 0.008$ | 29.00 (2.10) $H = 9.70; p = 0.008$ | 27.50 (2.57) $H = 9.70; p = 0.008$ |
| **$Wn^2$ (load volume in a set, kg) in anaerobic glycolysis energy supply mode** | | | |
| Hammer chest press | 474.40 (30.60) $H = 41.28; p < 0.001$ | 498.40 (25.80) $H = 41.28; p < 0.001$ | 388.40 (39.20) $H = 41.28; p < 0.001$ |
| Hammer lat pulldown | 368.00 (32.00) $H = 33.77; p < 0.001$ | 385.60 (24.00) $H = 33.77; p < 0.001$ | 350.80 (20.80) $H = 33.77; p < 0.001$ |
| Hammer triceps extension | 228.00 (23.00) $H = 9.70; p = 0.008$ | 232.00 (16.00) $H = 9.70; p = 0.008$ | 220.00 (21.60) $H = 9.70; p = 0.008$ |

**Notes.**
H, Kruskal–Wallis test criterion.

In each control exercise, elderly men had the lowest maximum strength development parameters after 12 months of bodybuilding training. Among individuals of late mature age (group B), the studied indicators exceeded those of group C by an average of 16.8% ($p < 0.05$). The 1RM results obtained during the control testing were 11.7% ($p < 0.05$) higher in group A representatives than in group C bodybuilders.

**Table 2 Results of changes in heart rate variability in the participants of the study groups during control tests (median, IQR), $n = 60$.**

| Groups | VCR indicators | | | | |
|---|---|---|---|---|---|
| | SDNN, ms | VLF, % | LF, % | HF, % | LF/HF |
| **Before exercise at rest** | | | | | |
| A | 54.30 (2.73) | 18.27 (3.62) | 60.66 (2.56) | 21.04 (0.97) | 2.89 (0.04) |
| B | 139.40 (6.57) | 6.44 (5.29) | 52.79 (2.65) | 40.47 (2.37) | 1.29 (0.04) |
| C | 42.20 (5.00) | 15.97 (5.09) | 64.33 (4.17) | 19.13 (1.29) | 3.34 (0.17) |
| **After exercise in the creatine phosphokinase energy supply mode** | | | | | |
| A | 33.40 (4.60)[*] | 52.68 (3.29)[*] | 41.03 (3.14)[*] | 6.29 (0.44)[*] | 6.52 (0.30)[*] |
| B | 196.70 (8.10)[*] | 17.52 (6.70)[*] | 38.49 (1.62)[*] | 44.43 (6.61)[*] | 0.87 (0.13)[*] |
| C | 79.65 (8.35)[*] | 18.40 (6.90)[*] | 42.42 (2.79)[*] | 39.08 (5.26)[*] | 1.09 (0.14)[*] |
| **After exercise in the anaerobic glycolysis energy supply mode** | | | | | |
| A | 67.77 (4.45)[*] | 25.42 (4.31)[*] | 48.03 (2.61)[*] | 26.43 (2.04)[*] | 1.83 (0.06)[*] |
| B | 156.30 (8.00)[*] | 7.30 (7.32) | 42.17 (2.10)[*] | 50.87 (5.70)[*] | 0.82 (0.07)[*] |
| C | 32.20 (5.07)[*] | 10.57 (3.88)[*] | 75.75 (3.12)[*] | 13.21 (1.56)[*] | 5.71 (0.55)[*] |

Notes.
[*]$p < .05$—compared to the results before exercise at rest.

Using an integral method of quantitative assessment of power loads (*Chernozub et al., 2018*), the indicators of the projectile working mass (m) and the load volume (Wn) were determined in the creatine phosphokinase and anaerobic glycolytic energy supply modes. In the creatine phosphokinase energy supply mode, all group representatives used 10.9% ($p < 0.05$) higher projectile working mass than in the other mode. At the same time, the Wn parameters were 80.9% ($p < 0.05$) higher in the anaerobic glycolysis energy supply mode.

Table 2 shows the results of vegetative regulation of heart rhythm and spectral characteristics of men of different age groups who are beginner bodybuilders. The studied indicators were assessed at rest and in response to exercise after control tests.

The HRV indicators at rest showed the following. The SDNN index in elderly men was 3.3 times lower than group B results. The difference between the representatives of early and late mature age was 2.5 times. This fact indicates the presence of a more intense regulation of heart rhythm in elderly men (group C) and early mature age (group A). The spectral analysis results showed that the autonomic balance in groups A and C was shifted towards sympathetic regulation. At the same time, group B representatives had a significantly higher proportion of HF ($p < 0.05$) than the others. The results of spectral analysis of the LF/HF index demonstrated the balance of mechanisms of vago-sympathetic tone in group B. Thus, spectral analysis of cardio intervals revealed a more advanced system of autonomic regulation in group B bodybuilders at rest.

The results obtained after study participants performed loads in the phosphocreatine energy supply mode showed significantly different changes in HRV indicators between the groups. The SDNN indicator after exercise demonstrated a significant increase in group B (+41.1%) and group C (+88.7%). In group A, an increase in heart rate regulation tension was observed, with SDNN decreasing by 38.5% ($p < 0.05$). In group A, there

**Table 3  Changes in blood biochemical parameters (creatine phosphokinase and lactate dehydrogenase enzymes, and cortisol and testosterone concentration) in study participants during control tests (median, IQR), $n = 60$.**

| Groups | Blood biochemical parameters | | | |
|---|---|---|---|---|
| | CPK, u/l | LDH, u/l | Cortisol, nmol/l | Testosterone, nmol/l |
| Before exercise at rest | | | | |
| A | 58.71 (6.29) | 222.06 (20.87) | 232.91 (28.01) | 16.42 (0.63) |
| B | 136.21 (10.60) | 213.44 (15.77) | 267.33 (14.82) | 18.72 (0.69) |
| C | 44.16 (1.19) | 219.62 (14.00) | 312.34 (14.79) | 8.92 (0.31) |
| After exercise in the creatine phosphokinase energy supply mode | | | | |
| A | 69.44 (2.97)[*] | 432.31 (22.24)[*] | 543.22 (34.54)[*] | 21.37 (1.59)[*] |
| B | 179.79 (12.10)[*] | 232.40 (13.01) | 309.53 (15.95)[*] | 23.25 (1.86)[*] |
| C | 132.72 (5.10)[*] | 229.35 (7.35) | 430.15 (23.25)[*] | 14.22 (0.96)[*] |
| After exercise in the anaerobic glycolysis energy supply mode | | | | |
| A | 62.01 (4.35)[*] | 422.39 (12.24)[*] | 183.95 (13.54)[*] | 18.47 (1.19)[*] |
| B | 149.88 (10.32)[*] | 291.26 (9.79)[*] | 395.42 (11.74)[*] | 21.28 (1.34)[*] |
| C | 66.42 (4.97)[*] | 312.37 (15.64)[*] | 273.58 (10.55)[*] | 10.35 (0.67)[*] |

Notes.

[*]$p < .05$—compared to the results before exercise at rest.

was a simultaneous decrease in LF (−19.6%) and HF (−14.7%) and a 2.2-fold increase in autonomic balance. Additionally, there was an enhancement of the central regulatory mechanism of the sinus rhythm, as indicated by a 34.3% increase in VLF. However, the HRV results in men from groups B and C showed a completely different pattern of changes. There was an increased influence of autonomic regulation (HF increased in group B by +3.4% and in group C by +19.6%) and a reduction in sympathetic tone (LF decreased in group B by −14.3% and in group C by −21.9%).

The analysis of the results in response to strength loads in the anaerobic glycolysis energy supply mode demonstrated various HRV indicator changes across the groups. The SDNN indicator significantly increased after the load in group A (+24.8%) and group B (+12.1%). Among elderly men (group C), a decrease in SDNN by 23.7% ($p < 0.05$) was observed, indicating an increase in heart rate regulation tension. In men from groups A and B, there was a significant increase in HF values alongside a decrease in LF values, reflecting an enhanced vagal influence on the sinoatrial node. In contrast, among elderly men, an increase in sympathetic tone (LF +11.4%) and a decrease in autonomic regulation influence (HF −5.9%) were noted. Significant differences were found between groups concerning changes in the autonomic balance index (LF/HF). The shift in autonomic balance toward parasympathetic regulation in response to such a stress stimulus indicates the activation of short-term adaptation mechanisms.

Table 3 shows the results of blood biochemical parameters (creatine phosphokinase, lactate dehydrogenase, cortisol, testosterone) of beginner bodybuilders of different age groups during the research. The studied indicators were assessed at rest and in response to control test exercises.

The analysis of the results obtained before exercise shows that the baseline biochemical blood parameters in the study participants corresponded to the reference values. However,

the creatine phosphokinase activity in the blood of group B representatives exceeded the results of the other two groups by 2–3 times. The basal testosterone concentration in the blood serum of the elderly men was almost 2 times lower than in groups A and B. At the same time, the cortisol concentration in the blood of group C participants was on average 17.6% ($p < 0.05$) higher than the results of other group representatives.

The biochemical monitoring of the participants' resistance levels to strength loads in the creatine phosphokinase energy supply mode had the following results. The greatest 3-fold increase in blood CPK in response to a stressful stimulus was found in elderly men (group C). The lowest increase in creatine phosphokinase activity by 18.3% ($p < 0.05$) was observed in the blood serum of group A representatives. At the same time, LDH (+94. 7%) and cortisol (+133.2%) in the blood serum of group A participants significantly increased. The LDH activity and cortisol concentration in the blood serum of groups B and C demonstrated an adequate physiological response to a stressful stimulus. The changes in the testosterone concentration in the blood of all study participants in response to acute exercise proved the mechanism of short-term adaptation. The main source during these loads in groups B and C representatives is creatine phosphate reserves. Due to insufficient creatine phosphate levels, group A participants additionally used muscle glycogen reserves during these loads, which increased LDH activity and cortisol concentration in the blood serum.

The laboratory control results showed that changes in some blood biochemical parameters in response to exercise in the anaerobic glycolysis mode differed from the previous ones. In these conditions, there was an increase in CK parameters by 10.0% ($p < 0.05$) in group B and by 50.4% ($p < 0.05$) in group C. LDH increased by 36.4% ($p < 0.05$) in group B and by 42.2% ($p < 0.05$) in group C. In group A, only LDH activity in the blood serum increased (+90.2%) compared to the state of rest. The cortisol concentration in groups A (−21.0%) and C (−12.4%) significantly decreased. A decrease in this hormone indicates a deficiency of muscle glycogen and the activation of compensatory reactions (gluconeogenesis) for the necessary energy supply for muscle activity. This fact highlights a low resistance level of groups A and C to loads in the anaerobic glycolysis energy supply mode.

## DISCUSSION

This study investigates the resistance levels of men from different age groups to anaerobic energy supply loads during their first 12 months of bodybuilding training. The research examined the nature of adaptive-compensatory responses in men of mature (first and second stages) and elderly age groups to acute strength loads. The findings indicate that beginner bodybuilders exhibit various resistance levels to loads in the creatine phosphokinase and anaerobic glycolysis energy supply modes. It was established that only men of late mature age showed no compensatory reactions to strength loads in both anaerobic energy supply modes.

The results of this study will contribute to improving the system of monitoring and managing training loads in bodybuilding, taking into account age-specific adaptation

processes to such stimuli. Furthermore, the findings will aid in addressing the challenge of identifying informative physiological and biochemical markers for evaluating resistance levels in men of different age groups to various loads during bodybuilding training.

The imperfection of the system for assessing the functional reserves of individuals across different age groups in bodybuilding, coupled with the lack of optimal mechanisms for adjusting loads, undermines the training process. This issue becomes particularly acute during the initial preparation stage. Implementing widely accepted training programs without evaluating the resistance levels of individuals from various age groups to physical stressors can lead to adaptation failures (*Baxter et al., 2024*; *Mastalerz et al., 2024*). Revising the current principles of training process modeling in bodybuilding, especially for middle-aged and elderly individuals, has recently caused significant debate (*Kraková et al., 2023*; *Lixandrão et al., 2024*). Even at the initial preparation stage, it is crucial to develop training models that consider the adaptive-compensatory responses of individuals of different ages to anaerobic loads (*Chernozub et al., 2023*).

Finding an optimal mechanism for engaging agonist and synergist muscles while simultaneously reducing the load on stabilizing muscle groups by adjusting the kinematic characteristics of exercise techniques is a pressing challenge (*Rukstela et al., 2023*; *Barbosa et al., 2024*). Determining the optimal range of physiological and biochemical indicators to assess adaptive reserves and monitor compensatory responses in different age groups within bodybuilding is a current scientific priority (*Chernozub et al., 2020*; *Güngör et al., 2024*).

Simultaneous use of heart rate variability (HRV) and blood biochemical markers (enzymes, hormones) as indicators for assessing the body's resistance to anaerobic loads is one of the ways to address this issue (*Sellami et al., 2018*; *Marasingha-Arachchige, Alcaraz & Chunga, 2022*; *Rial-Vázquez et al., 2023*). However, leading researchers have only partially focused on the problem of resistance to anaerobic energy loads in men of different age groups within bodybuilding. This may be because most studies have examined adaptation processes in other age groups and preparation stages in bodybuilding.

The baseline HRV results indicate that beginner bodybuilders in the elderly and early mature age groups experience the highest tension in heart rhythm regulation. The spectral analysis showed that this age category shifted autonomic balance toward sympathetic regulation. The basal creatine phosphokinase level in the blood of late mature age men was 2–3 times higher than in the other age groups. Among elderly men at rest, the lowest basal testosterone levels and the highest cortisol concentrations in blood serum were observed. These findings suggest a possible decrease in the resistance of groups A and C to stress stimuli due to adaptation failure caused by overloading (*Chycki et al., 2024*; *Jin et al., 2024*).

Following exercise in the creatine-phosphokinase energy supply mode, heart rhythm regulation tension and central control activation increased in men of the early mature age group. Simultaneously, despite minimal increases in creatine phosphokinase, a significant rise in LDH and cortisol levels, nearing the upper limit of normal, was observed in this group. These changes indicate insufficient creatine phosphate reserves, necessitating the use of muscle glycogen to counteract loads in this anaerobic mode (*Rial-Vázquez et al., 2023*; *Gambassi et al., 2024*). The substantial increase in blood cortisol levels under these

conditions suggests compensatory responses to the stress stimulus (*Athanasiou, Bogdanis & Mastorakos, 2023*). Under similar conditions, members of other groups exhibited enhanced autonomic regulation and reduced sympathetic tone. The biochemical blood markers in this study reflect the positive outcomes of effective short-term adaptation processes. These findings corroborate the results of several leading researchers studying the peculiarities of adaptive-compensatory reactions under strength load conditions (*Lin et al., 2022*; *Mastalerz et al., 2024*).

The decrease in cortisol concentration and the simultaneous significant increase in LDH levels in the blood of groups A and C in response to anaerobic glycolysis energy supply mode loads indicate compensatory reactions of the body. This compensatory response to anaerobic physical activity suggests a deficiency in the required muscle glycogen reserves in the working muscles (*Athanasiou, Bogdanis & Mastorakos, 2023*; *Chernozub et al., 2023*). The lack of sufficient muscle glycogen reserves under strength load conditions in anaerobic glycolysis energy supply mode activates gluconeogenesis processes to stabilize the energy supply system (*Kotikangas et al., 2022*). However, low reserves of muscle glycogen and creatine phosphokinase (CPK) are associated with insufficient rates of muscle mass growth through hypertrophy during long-term adaptation (*Carvalho et al., 2022*; *Baxter et al., 2024*). One of the reasons for this issue is the absence of an integrated system for monitoring and adjusting training loads, taking into account the body's resistance to stimuli and age-related characteristics.

## CONCLUSIONS

The research findings highlight the necessity of developing new mechanisms for adjusting training loads for mature and elderly individuals starting bodybuilding. When designing training regimes, it is essential to consider baseline morphometric parameters, strength capabilities, and the body's adaptive reserves. Using HRV indicators and blood biochemical markers (lactate dehydrogenase, creatine phosphokinase, cortisol) has been substantiated as an effective tool for evaluating adaptive-compensatory responses to stimuli. These markers allow for a precise assessment of the resistance level of beginner bodybuilders in mature and elderly age groups to anaerobic energy supply loads. The study showed that only men in the late mature age group demonstrated more advanced autonomic heart rhythm regulation after 12 months of bodybuilding. This age group also exhibits high resistance to loads in anaerobic energy supply modes (creatine phosphokinase and glycolytic). Using an integrated method for quantitatively assessing strength loads in the development of functional tests creates optimal conditions for evaluating the resistance levels of different age groups to stress stimuli.

## ACKNOWLEDGEMENTS

The article is a part of the planned scientific work "Development and implementation of innovative technologies and correction of the functional state of a person during physical activity in sports and physical therapy", (state registration number 0117U007145) and Ministry of Education and Science of Ukraine (0118U000809).

### Funding

The authors received no funding for this work.

### Competing Interests

The authors declare there are no competing interests.

### Author Contributions

- Andrii Chernozub conceived and designed the experiments, performed the experiments, analyzed the data, prepared figures and/or tables, authored or reviewed drafts of the article, and approved the final draft.
- Alla Aloshyna conceived and designed the experiments, performed the experiments, prepared figures and/or tables, authored or reviewed drafts of the article, and approved the final draft.
- Georgiy Korobeynikov conceived and designed the experiments, performed the experiments, analyzed the data, prepared figures and/or tables, authored or reviewed drafts of the article, and approved the final draft.
- Vadym Koval conceived and designed the experiments, performed the experiments, prepared figures and/or tables, authored or reviewed drafts of the article, and approved the final draft.
- Yurii Havrylov conceived and designed the experiments, performed the experiments, prepared figures and/or tables, authored or reviewed drafts of the article, and approved the final draft.
- Liudmyla Sherstiuk conceived and designed the experiments, performed the experiments, analyzed the data, prepared figures and/or tables, authored or reviewed drafts of the article, and approved the final draft.
- Vladimir Potop conceived and designed the experiments, performed the experiments, analyzed the data, prepared figures and/or tables, authored or reviewed drafts of the article, and approved the final draft.
- Consuela Andreea Timnea-Florescu conceived and designed the experiments, performed the experiments, prepared figures and/or tables, authored or reviewed drafts of the article, and approved the final draft.
- Olivia Carmen Timnea conceived and designed the experiments, performed the experiments, analyzed the data, prepared figures and/or tables, authored or reviewed drafts of the article, and approved the final draft.

### Human Ethics

The following information was supplied relating to ethical approvals (*i.e.*, approving body and any reference numbers):

The algorithm, structure, and methods of the study were approved by the ethical committee for biomedical research of Lesya Ukrainka Volyn National University (Minutes No. 023384 dated September 19, 2024).

## Data Availability

Raw data is available as a Supplemental File.

## Supplemental Information

Supplemental information for this article can be found online at http://dx.doi.org/10.7717/peerj.19844#supplemental-information.

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
