# Peer review of "Resistance of mature and elderly bodybuilders to anaerobic energy supply load"

_PeerJ, doi:10.7717/peerj.19844_

## Round 0.1 · original submission · Major Revisions

**Language Note:** The review process has identified that the English language must be improved. PeerJ can provide language editing services - please contact us at [email protected] for pricing (be sure to provide your manuscript number and title). Alternatively, you should make your own arrangements to improve the language quality and provide details in your response letter. – PeerJ Staff

Reviewer 1 ·

Basic reporting

The manuscript currently contains excessively complex language that decreases readability. For example, p. 3, lines 77–78 includes the phrase "adaptive-compensatory reactions of individuals to various modes of anaerobic energy supply," which seems dense and rather nebulous. Perhaps something like "adaptive responses to anaerobic exercise modes" would work instead? Also, please include a brief definition for "adaptive-compensatory reactions." Similarly, p. 13, lines 310–311 includes the sentence "The baseline HRV results indicate that beginner bodybuilders in the elderly and early mature age groups experience the highest tension in heart rhythm regulation," which is hard to follow. Multiple examples like this can be found within the manuscript, and it would be very helpful to modify the language throughout for clarity, perhaps by asking someone who is a native English speaker and kinesiology professional to review it.

The manuscript certainly references relevant studies, but it would be particularly helpful to include recent research on resistance training adaptations in older adults. For example, the introduction might benefit from discussing findings on the physiological and hormonal adaptations to resistance training in older populations.

The tables (especially table 2) seem dense and challenging to interpret, perhaps because the manuscript does not currently adequately contextualize the data. Visual summaries (bar graphs, trend lines, etc.) would improve clarity and accessibility. The table legends would also benefit from including explicit definitions for key metrics like SDNN and LF/HF for readers who are less familiar with the topic.

Experimental design

The research question is not clearly stated, which makes it hard to understand the study aim. For example, p. 3, line 94 mentions that the purpose of the study is "to define the resistance level of beginner bodybuilders," which lacks specificity.

The study does not provide enough detail on how rigorous technical and ethical standards were upheld. While ethical approval is mentioned (p. 7, line 107), there is no explanation of how participant safety was monitored, especially for older adults performing potentially strenuous anaerobic exercises.

The methods don’t appear to be described with enough detail to permit full replication. For example, p. 9, lines 123–129 describes load volume calculations and includes formulas, but doesn’t account for adjustments related to age or fitness levels. As another example, the procedures for heart rate variability measurements are described on p. 9, lines 130–135), but omit details on how external factors, like environmental conditions, were controlled. Step-by-step descriptions of key procedures and a discussion of measures taken to ensure consistent testing conditions would be helpful.

Validity of the findings

All underlying data are provided, but they are not adequately contextualized to assess robustness or control for confounding factors. For example, LDH and cortisol levels are presented in Table 3, but baseline variability and potential influences such as diet or medication are not addressed. It would be helpful to make sure that confounding factors are controlled and to describe these controls in the methods section.

The conclusions occasionally overstate the findings. For example, p. 14, line 352 states that "only men in the late mature age group demonstrated advanced autonomic regulation," which seems to overlook variability within groups. It would also be helpful to include limitations of the research.

The study does not appear to provide actionable recommendations based on the results. For example, while highlighting differences in physiological responses across age groups, would it be possible to translate these findings into practical guidelines for trainers or clinicians? Perhaps you could discuss specific applications of the findings, like tailored training programs or monitoring strategies for beginner bodybuilders in older age groups.

Reviewer 2 ·

Basic reporting

Why did they name the participants Physiotourists?

The term anaerobic glucose is sometimes used, but is there such a thing as aerobic glucose?

What was the alternative hypothesis of the study?

Experimental design

Why is it relevant to analyze biochemical and cardiovascular factors for load control specifically in strength training?

How important is it to know functional reserves for bodybuilders?

How long should I rest before collecting heart rate variability?

What criteria was used to choose the number of participants?

Validity of the findings

What do the results bring that is different for the assessment and prescription of strength training?

---

## Round 0.2 · accepted · Accept

The authors have addressed all of the reviewers' comments, and now this manuscript is ready for publication.

Reviewer 1 ·

Basic reporting

-

Experimental design

-

Validity of the findings

-